# eHealth and mHealth in Chronic Diseases—Identification of Barriers, Existing Solutions, and Promoters Based on a Survey of EU Stakeholders Involved in Regions4PerMed (H2020)

**DOI:** 10.3390/jpm12030467

**Published:** 2022-03-15

**Authors:** Dorota Stefanicka-Wojtas, Donata Kurpas

**Affiliations:** 1Clinical Trial’s Department, Wroclaw Medical University, 50-556 Wroclaw, Poland; 2Family Medicine Department, Wroclaw Medical University, 51-141 Wroclaw, Poland; donata.kurpas@umw.edu.pl

**Keywords:** mHealth, eHealth, chronic diseases, interregional cooperation, barriers, facilitators, healthcare systems

## Abstract

Background: In recent years, rapid population ageing has become a worldwide phenomenon. Both electronic health services (eHealth) and mobile health services (mHealth) are becoming important components of healthcare delivery. The market for mHealth is growing extremely fast. However, despite the increasing investment and interest in eHealth, several challenges still need to be overcome to enable broader and more systematic implementation of ICT in healthcare. Methods: This study presents data from the survey “Barriers and facilitators of Personalised Medicine implementation- qualitative study under Regions4PerMed (H2020) project”. In addition, this paper discusses the results of the conference, Health Technology in Connected & Integrated Care, held under the Horizon 2020 project and interregional coordination for a fast and deep uptake of personalised health (Regions4Permed) (July 2020—online conference). The above sections were preceded by an analysis of existing articles. Results: The data obtained from the surveys show that the main barriers to the adoption of eHealth and mHealth are the lack of skills of seniors, but also the lack of user-friendly technology and a simple user interface. Access to individual data while ensuring its security and the lack of digitisation of medical data are also serious issues. In addition, medical digital solutions are overly fragmented due to national legislations that deviate from the General Data Protection Regulation. Conclusions: By using technological solutions, it is possible to improve diagnosis and treatment decisions, and better adapt treatment and reduce its duration and cost. However, there are still barriers to the development of eHealth. Clear recommendations for implementation are needed to enable further development of personalised eHealth and mHealth solutions

## 1. Introduction

Electronic health (eHealth) and mobile health (mHealth) services are becoming indispensable components of healthcare, covering a wide range of health services, from electronic prescriptions and medical records to patient communications to improve patient compliance [1].

The future of European healthcare systems and the implementation of personalised medicine in clinical practice require smart planning and a structured approach to ensure quality and long-term sustainability. 

To this end, attention should be focused, in particular, on technologies and ICT that can deliver the desired outcomes for reorienting personalised and patient-centred healthcare systems.

Quality of care must be the focus of attention for politicians and decision-makers in all regions of Europe. Health services need to prioritise people with multiple chronic conditions (comorbidity), and it seems that this can be achieved through an integrated and patient-centred approach that allows the needs of patients to be adequately met [2]. The authors, e.g., Talboom—Kamp et al. (2016), emphasise that the pressure to implement self-management through eHealth is immense. The reason for this is the number of people with chronic diseases and comorbidity, which is rapidly increasing due to rapid ageing and higher life expectancy of the population [3].

The eHealth market is growing extremely rapidly and is expected to increase from $45 billion in 2020 to $194 billion in 2027, with a CAGR of 23% in the forecast year from 2021 to 2027 [4]. However, despite the growing investment and interest in eHealth, several challenges still need to be overcome to enable a broader and more systematic implementation of ICT in healthcare. Healthcare services and systems must become more resilient, effective, equitable, accessible, sustainable, and comprehensive. Transformation and adaptation require a digitally oriented mindset [3]. Foster and Sethares (2014) point out that the use of telehealth by older people will increase, but that effective adoption of such interventions can only be successful if the patient’s perspective is at the forefront [5].

Pilot projects (including the one presented below) are being conducted around the world, and areas of opportunity are being identified that could potentially have global impact. Barriers to large-scale implementation such as standards, security, and interoperability are also being identified [6]. As Kampmeijer et al. (2016) noted, the successful use of eHealth/mHealth tools in health promotion programs for older people is highly dependent on the motivation and support they receive in using eHealth and mHealth tools [7]. It is also worth noting that according to Alwashmi et al. (2019), mHealth could increase patient autonomy by stimulating patient empowerment and motivation [8].

## 2. Purpose of the Study

The purpose of this paper is to identify the main barriers, facilitators, and interventions related to eHealth and mHealth services in chronic diseases, and to present the state of implementation of eHealth and mHealth services in daily practice by describing the research findings and existing gaps.

## 3. Materials and Methods

### 3.1. Study Design

This study presents data from the survey, “Barriers and facilitators of Personalised Medicine implementation- qualitative study under Regions4PerMed (H2020) project”. The survey included: general information on gender and nationality, two questions on individual experiences with barriers and facilitators of personalised medicine implementation, and five questions about the implementation of personalised medicine itself. The qualitative study—survey research—is in progress. This article shows only results that are relevant to the goal of the study.

In addition, this paper discusses the results of the conference, Health Technology in Connected & Integrated Care, held as part of the Horizon 2020 project and interregional coordination for a fast and deep uptake of personalised health (Regions4Permed) (July 2020—online conference). The conference explored the impact of new regulations and what solutions can be adopted at the regional level:Training for healthcare professionals: health technologies are often beneficial to patients, but health professionals often encounter difficulties in using the equipment associated with these technologies, which can increase the risk of accidents;Patient engagement: there is growing evidence that increased patient awareness leads to better care and treatment adherence, as well as improved health and well-being;Economic impact of health technologies: critical to the market access of health technologies and the implementation of personalised medicine is the valuation of health technologies, which needs to be discussed to assess market barriers and identify potential solutions [2].

The above sections were preceded by an analysis of existing articles. Using the desk research method, it was possible to extract them in the form of published materials, online databases, available literature, and documents. Systematic reviews of barriers and facilitators of eHealth and mHealth published in Medline and Academic Search Ultimate, between January 2015 and December 2021, were evaluated and interpreted according to PRISMA recommendations. The following keywords were used throughout the research process: mHealth, eHealth, barriers, facilitators, challenges, chronic diseases.

### 3.2. Setting

The article is based on the results of online surveys and the summary of the two-day conference, Health Technology in Connected & Integrated Care, held in July 2020 (online). Three sessions and one on-site day were held during the conference.

### 3.3. Participants

Survey: The 69 online surveys from 19 countries (Lithuania, Germany, Italy, France, Kazakhstan, Poland, Spain, Netherlands, Denmark, Portugal, Turkey, Latvia, Greece, Canada, Ukraine, Belgium, Estonia, Romania, and Sweden) were conducted. The participants of the survey included 33 women and 36 men, aged 24–74 years. The participants are experts in the field of eHealth, mobile health services, and personalised medicine. They are associated with research institutes, private founders, and local and national governments.

The questionnaires were sent to the stakeholders of the Regions4PerMed (interregional coordination for a fast and deep uptake of personalised health) project: presenters at conferences and workshops, and participants in these events. This methodology allows for analysis of experiences related to barriers and facilitators to implementation based on evidence from existing implementations of innovative eHealth interventions at micro-, meso-, and macro-regional levels within health and social care systems, particularly related to chronic disease management.

At the Health Technology in Connected & Integrated Care conference, 16 speakers, and 3 site visits were conducted. The conference was attended by 86 registered participants, and the average time spent in the room was 170.5 min.

### 3.4. Data Sources

This study represents an analysis of data obtained from existing articles [7,8,9,10,11,12,13,14,15,16,17,18,19,20,21,22,23,24,25,26,27,28], online surveys (data from the survey “Barriers and facilitators of personalised medicine implementation- qualitative study under Regions4PerMed (H2020) project”, author of the survey Dorota Stefanicka—Wojtas), and results of the Health Technology in Connected & Integrated Care conference [2]. 

### 3.5. Study Size

Between July 2020 and January 2022, 69 surveys were conducted. The study used a series of structured questions in an online questionnaire (using Google Forms). Data were collected, stored in a database, and included in the study. Conference participants were experts from academia and industry, and representatives of regional and governmental health policy institutions from a variety of EU countries. The number of conference participants was: 86 registered participants and 19 speakers from 17 countries

### 3.6. Variables

#### 3.6.1. Quantitative Variables

The survey included questions on quantitative variables and general information such as gender and nationality.

#### 3.6.2. Qualitative Variables

The survey included questions on unstructured qualitative variables, including two questions on individual experiences related to barriers and facilitators to implementing personal medicine, and five questions on the implementation of this concept. This article shows only results relevant to the goal of the study. Example questions:
“In your opinion, what are the most important facilitators and barriers to public use of personalised medicine? What do the barriers/facilitators relate to (types of barriers identified, e.g., health care system, government, and PM users)? Please list and explain them briefly.”

### 3.7. Ethics Approval

The survey was ethically approved by the Bioethics Committee of the Medical University of Wroclaw under the number KB0450/2020. The participants of the conference and workshops were asked for their consent to record and use their statements; the permission was granted.

## 4. Results

### 4.1. Systematic Reviews of Barriers and Facilitators of eHealth and mHealth

There were 5337 records identified using Medline and Academic Search Ultimate (Figure 1). The keywords used throughout the research process were: m-health or e-health, barriers and facilitators, and challenges and chronic diseases. The literature search identified a total of 5337 potentially relevant studies. Studies were eligible if they were published in peer-reviewed academic journals, written in English, and full text was available. Of the remaining 1504 records, 1187 were excluded after title and abstract assessment, and 297 were excluded after full-text analysis. The remaining 22 studies were analysed. The following Table 1 and Table 2 shows the overview of the results. 

For the record identified through Medline and Academic Search Ultimate databases (articles published between 2015–2022), *n* = 5337.

For the dataset excluded (abstracts only, published in peer-reviewed academic journals, in English), *n* = 3833.

Full text articles screened for eligibility *n* = 1504.

Exclusion of articles after reading title and abstracts *n* = 1187.

Exclusion of articles after full text analysis *n* = 297.

Full text studies included in the qualitative synthesis *n* = 22.

### 4.2. Participants and Descriptive Data—Survey

The next step was an online survey of women and men associated with research institutions, private funders, policy makers, policy advisors, project managers, researchers, patient advocates, physicians, scientific officers, and health science consultants.

The research conducted made it possible to identify the barriers and facilitators observed in European countries. The following Table 3 shows the expected barriers and facilitators to Quadruple Aim implementation (data from surveys). We also include additional conclusions from the surveys.

Identifying these barriers and facilitators will allow for the development of recommendations to reduce barriers to the implementation of eHealth and mHealth in chronic diseases. 

As mentioned in the survey, the lack of access to individual data, while ensuring their security, and the lack of digitialisation of medical data, are serious issues [AR_IT]. Another obstacle is the regional fragmentation of healthcare, which makes it difficult to share data, or implement PM measures [MN_ES] and the not very widespread use of data and supporting data (apart from classical laboratory tests) [SB_DE]. Data collected in the surveys suggest that the next main barriers to adoption of eHealth and mobile health services are seniors’ lack of knowledge [AT_IT], but also the lack of user-friendly technology [EB_IT] and a simple interface [AHA_TR].

Surveys show that both facilitators and barriers are related to the management, organisation, and functioning of (public) healthcare systems. PM (personalised medicine) solutions should work in synergy with the system to increase the value of healthcare to patients. Too often, PMs create specific/individual silos that are not aligned with the system and only exacerbate the fragmentation of the system. As a result, they very soon become classified as “gimmicks” and disappear [TP_PL]. Progress in PM can only be achieved through increased awareness among patients and caregivers, (HCPs are already aware of this) and policy makers (regulators, policy makers) who need to understand the potential and pathway of PM. There is a need for training and conferences for the general public, as well as roundtables and workshops to promote dialog between agencies/regulators/policy makers and clinicians/researchers in the field of PM [MV_IT]. While patients are optimistic about personalised treatment, they are not familiar with the principles of personalised medicine. Users of personalised medicine can be considered as facilitators, as they can contribute to the dissemination of the principles and their personal experiences [MD_GR].

An information campaign on the scale of pharmaceutical or automobile advertising should be undertaken. Specifically, public resources run by professionals to provide truthful, verifiable, and understandable information to the public. This should be supplemented by podcasts, TV, and YouTube documentaries, and by prominently placed and supported web content [MW _DE]. Patient associations should also be established. They can represent patients and citizens [MN_ES].

Meaningful integration of PM requires several paradigm shifts in both public option and decision making. Citizens need to be much better informed about personalised medicine, i.e., healthy people and NOT patients; this is far too late, because more than half of PM options are already over by the time one becomes a patient [MW_DE]. Like any radical change in an industry that has never focused on individualising citizens, the introduction of personalised medicine is likely to be met with resistance. This could be related to the unwillingness of health professionals to change their current practices and the unwillingness of the industry to make large investments until it is absolutely certain of a payoff [MA_IT].

Respondents emphasised that there are many facilitators for eHealth and mobile health adoption in Europe. One of these is communication and education of citizens about the benefits of these solutions [KR_EE]. In some countries, personalised data already serve as a decision-making tool for personalised diagnosis and treatment [GK-J_DE]. As interviewees emphasised, healthcare providers can ensure direct contact with patients and better explain the benefits of eHealth and mobile health services [MCN_ES], and teach patients how to take care of their mental and physical health.

Personalised medicine also requires a different approach to patients by health professionals [KZ_PL], who can engage directly with patients and better explain the benefits of PM treatment to them [MN _ES]. It is pointed out that there is a lack of training of health professionals and the lack of investment in health care [MFG_ES].

It is noted that there is a lack of infrastructures and homogeneous regulations that allow stakeholders to practice personalised medicine [MS_DE]. Medical digital solutions are too fragmented due to national legislations that deviate from GDPR/national assessment [JT_FR]. A centralised assessment system and transparency between reimbursement rates of national healthcare systems [JT_FR] are also important.

Barriers often arise from regulations (e.g., privacy laws that hinder the use/reuse of personalised information, or regulatory issues that limit the ability to prescribe innovative and personalised methods because the process to get them reimbursed by the public health system can be very slow) [MV_IT].

### 4.3. Participants and Descriptive Data—Conference

Many pilot eHealth and mHealth projects in chronic diseases are being conducted worldwide. Based on the Regions4PerMed project, the results of the conferences, workshops, and best practices brochures of the Regions4PerMed project for selected existing solutions in the field of personalised medicine have been compiled in a Table 4.

## 5. Discussion

As noted by Wilson et al. (2021), rapid population ageing has become a global phenomenon. In 2018, the number of seniors exceeded the number of children for the first time in history. From the data they presented, they will account for 22% of the world’s population in 2050. This is one of the most important reasons to ensure adequate planning and delivery of health and support services [9].

Older people require many more multicomponent interventions; they tend to be more vulnerable not only to chronic diseases but also to side effects of their treatment, to which patients may respond very differently when these effects are influenced by age. Limited income or insurance coverage, limited mobility, disability, rural or remote location, and negative self-perception of ageing (associated with lower health-related quality of life) should also be considered [3,9,10,33]. Therefore, both electronic health services (eHealth) and mobile health services (mHealth) are becoming indispensable components of healthcare. eHealth/mHealth encompass a wide range of healthcare services, from electronic prescribing and access to medical records to text messages reminding patients to take their medications. Therefore, eHealth and mHealth are becoming important components of healthcare delivery [34].

### 5.1. Chronic Diseases Management

Adherence to chronic disease management is critical for better health outcomes, quality of life, and cost-effective healthcare. Mobile technologies are increasingly being used in health care and public health practice (mHealth) for patient communication, monitoring and education, and facilitating adherence to chronic disease management [35].

Mobile health technology (mHealth) supports medical practice. It has proven useful in daily self-management of chronic diseases by patients themselves or in remote medical management. The processing, sensing, and communication capabilities of mobile devices such as smartphone applications, web-based technologies, telecommunication services, social media, and wearable technologies are becoming increasingly popular. This has led to their use as the main technology for providing comprehensive health services [36,37].

In recent years this has led to a strong focus on promoting “self-management” among chronically ill patients. Although patients who are more knowledgeable about their disease, health, and lifestyle (especially when they are responsible for managing their own health and disease) have better experiences and health outcomes and often use fewer health resources, it is noted that limited knowledge about self-management can further limit health behaviours in patients with chronic disease [3,11]. To overcome this barrier, lifestyle changes and an approach called “self-management” of chronic disease are needed. It is important to empower patients to actively participate in health management with a focus on complete physical well-being. This includes introducing innovative approaches to existing health care such as medical management and changing, maintaining, and creating meaningful behaviours [11,38].

### 5.2. System for the Collection of Medical Data in Chronic Diseases

One of the main objectives in electronic health services (eHealth) and mobile health services (mHealth) is to use medical data collection systems in chronic diseases [34]. 

Another goal is to increase knowledge and strengthen the citizens’ and community’s participation in the surveillance system. Inadequate access to medical data and a lack of confidence in its quality are the main causes of underutilisation of ICT [34].

It is very important to link data between different local data centres, in order to make treatment decisions [2]. Both evidence and data infrastructures capable of collecting and analysing data are needed. For this reason, more investments should be made in these areas [2].

Integration is needed in defining new standards for topics such as cross-platform authentication and data exchange. Standardisation in healthcare services is an important prerequisite for improving patient care through the use of modern technologies. The development of standards is very important to ensure the interoperability of information systems, i.e., to enable them to communicate with each other and to enable eHealth projects around the world. Therefore, it is necessary to develop new and comprehensive standards [2]. The highly personalised data captured by digital health technologies can improve the relationship with patient outcomes and treatment adherence, and enable broader use of value-based care models. They enable both informed decision making and the delivery of personalised care [39,40].

### 5.3. Use of eHealth and mHealth by Older People

There are still barriers to the adoption and use of eHealth by older people.

Nevertheless, it is observed that the use of information and communication technology by the elderly is increasing and is considered positive and important in their daily lives. This considerable potential makes it possible to better meet the health needs of older people. The design of digital health services must be based on the specific needs of older people. The main barriers related to the functionality of eHealth platforms and related issues are the lack of an age-appropriate interface, small screens, small texts, etc. However, there are other age-related barriers that affect the usability of mHealth. These include lack of knowledge about how to use mHealth, high cost of new technologies, and limited/fixed income. Very often, mobile technologies are too difficult to use. Seniors can be overwhelmed by new information and alerts. In summary, when developing eHealth services for seniors, it is very important to include features such as audio feedback, a large text size, and a notification system that allows users to choose how and when they are notified so they can engage with the platforms that work best for them. Increasing this awareness influences the detection and effective management of chronic diseases and the reduction of their prevalence in the population [10,12,41,42].

Increasing awareness influences the detection and effective management of chronic diseases and the reduction of their prevalence in the general population. The initiative focuses on older people and their needs in terms of prevention and management of frailty. Health systems integration is multidimensional and complex, involving multiple stakeholders. Future challenges and vulnerable health systems require smart solutions that support the continuum of care for frail patients [2].

### 5.4. Patient Engagement in eHealth and mHealth

There is growing evidence that better-informed patients improve their self-care and medication adherence and enhance their health and well-being. The ability of citizens to access data is also considered important for reasons of improving disease management as a new form of patient engagement and empowerment [2]. 

For consumers who use mobile devices to access their medical records through online portals, a good experience is more than a matter of convenience. A recent study shows that it can lead to patients staying in touch with their primary care physician more regularly and requiring fewer hospitalisations. A Kaiser Permanente study that pooled diabetes patients from 2006 to 2007 [43], and patients with multiple chronic conditions, confirmed that those who connected with health resources via smartphones, tablets, or computers had better outcomes [44]. Similarly, research by Lee et al. (2018) on the impact of mHealth app feature usage patterns on user engagement showed that users used the app the most at the time of launch, and their usage gradually declined over time. The research suggests that frequent and regular use of the self-monitoring function significantly reduces the likelihood of abandoning the app. Thus, sustained use of mHealth apps is closely associated with regular use of the self-monitoring function [45].

In Poland, for example, the National Health Fund should and will continue to develop the potential for providing tools for patients and healthcare providers to facilitate patient–doctor communication, by introducing a new type of service based on telemedicine and telehealth diagnostics that enables patients to manage their own health. The primary care physician’s team should take care of them because they have a better and closer contact with patients and know their needs [2].

### 5.5. eHealth and mHealth—Quality of Life

There is an urgent need to transform health care systems as the population ages rapidly and the prevalence of chronic disease and comorbidity continues to increase. Therefore, it is critical to adapt the way patients and medical professionals communicate and collaborate to promote health. This is the only way to meet future expectations for high-quality, patient-centred care [39]. Healthcare services need to prioritise care for people with multiple chronic conditions (multimorbidity) and this seems to be best accomplished through integrated and patient-centred approaches to adequately meet patients’ needs. Despite the increasing investment and interest in eHealth, there are still some challenges to overcome to enable wider and more systematic adoption of ICT in healthcare. Health services and systems must become more resilient, effective, equitable, accessible, sustainable, and comprehensive. Transformation and adaptation require a digitally oriented mindset [2].

The application of eHealth solutions can provide chronically ill patients with high-quality care that satisfies both patients and healthcare professionals, while reducing healthcare consumption and costs [3].

### 5.6. Integrated Care Policies

The main challenge for national and regional authorities is to coordinate regional PM policies and innovation programs, to improve system integration and patient management, and to accelerate the use of PM for citizens and patients [46]. Also, very important at this moment is the lack of an eHealth policies that would promote effective strategies for adoption by clinicians [13].

The surveys also showed that there is a great need for mutual recognition for medical digital solutions published in other EU Member States. 

In addition, the conference highlighted that eHealth and mHealth also require not only cross-border and interdisciplinary collaboration in chronic disease management, but also stakeholder engagement [2].

### 5.7. Healthcare Providers in Chronic Diseases

Citing the need to find innovative solutions to meet the needs of citizens, new technologies are expected to transform health care delivery. Any holistic method of delivering health services to the public should improve health care by optimising physician–patient relationships. The improvement of health care should be achieved through the introduction of user-friendly technological tools that are easy to use for patients of all ages [47].

It was also pointed out that developing the potential to provide patients and health care providers with proven tools will facilitate the patient’s communication with the physician by introducing a new type of service based on telemedicine and telehealth diagnostics, while empowering the patient to manage his or her own health; that technology can help detect emergencies and assist healthcare professionals by providing them with early access to health information and recommendations [2].

## 6. Limitations of the Study

This review has some limitations. The search was limited to articles published in English, or only English-language papers were included. The exclusion of articles in other languages may have limited access to studies with significant results related to our objective. To overcome this limitation, PubMed/MEDLINE and Academic Search Ultimate databases were searched for studies published in English between 2020 and 2021. 

Survey data and conference data also have limitations. The number of participants may be considered an insufficient sample size for statistical measurement. The content of the questions may also be considered a limitation of the study. However, it is important to note that the questions were approved by the supervisor (professor with experience in qualitative and quantitative research) and that the questionnaires were sent to the stakeholders of Regions4PerMed: the lecturers of the conferences and workshops, and the participants of these events. This methodology allows for analysis of experiences related to barriers and facilitators to implementation, based on evidence from existing implementations of innovative eHealth interventions at the micro-, meso-, and macroregional levels within health and social systems, particularly related to chronic disease management. Conference participants included experts from academia and industry, as well as representatives of regional and governmental health policy institutions from a variety of EU countries.

## 7. Conclusions

The increasing burden of chronic disease requires innovative approaches to chronic disease prevention and management. Much has already been done to improve the quality of life in chronic diseases through personalised medicine. The use of technological solutions improves diagnostics and the treatment decision-making process, enables better customisation of treatment, and reduces its time and cost. eHealth facilitates diagnostics, prevention, and treatment. The concept of eHealth aims to break down barriers so that healthcare providers (government agencies, hospitals) can work more closely together.

However, there are still barriers to the development of eHealth, such as inconsistent legislation. The main obstacles to collaboration with government agencies are bureaucracy and lengthy legislative procedures. It should also be remembered that the computer level in medical institutions is low and there is a lack of well-trained personnel and coherent global platform. To enable further development of eHealth and mHealth electronic medical records and medical event registration are needed.

In summary, medical technologies in the areas of prevention, diagnosis, treatment, and rehabilitation occupy an important place in home care, ambulatory care, and hospital departments. Medical technologies for home care can help modulate home care and make it more effective in terms of cost and treatment time. With the advent of new technologies that increase the efficiency of health care delivery, it is critical to improve the population’s competence in using these technologies.

## Figures and Tables

**Figure 1 jpm-12-00467-f001:**
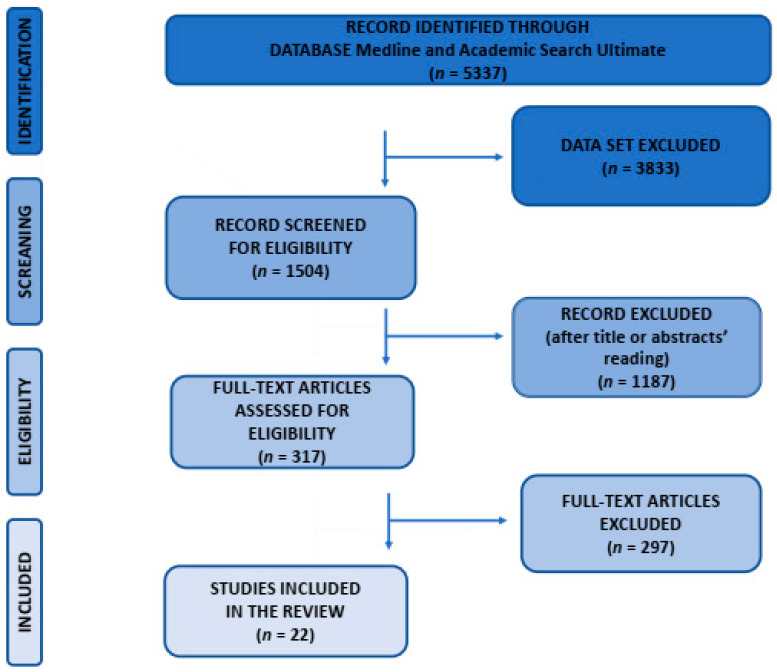
Flow diagram of identified and included records, according to PRISMA guidelines.

**Table 1 jpm-12-00467-t001:** Overview of the results—barriers.

	**Barriers to eHealth and mHealth Implementation in Chronic Diseases**	**References**
Individual	- lack of awareness of eHealth and mHealth services	[7,9,18,21,22,24,27]
	- lack of experience and knowledge	[10,11,12,14,17,18,20,22,25]
	- lack of necessary equipment	[8,12,20,24]
	- lack of motivation	[10,20,23,24,25]
	- cost of new technology, lack of access to electronic devices	[7,8,9,12,17,20,24]
	- lack of or limited trainings	[7,9,17,21,26]
Technological	- user-friendly technical tools	[7,9,10,12,14,18,22,23,24,28]
	- safety precautions	[11,15,18,19,20,21,23,24,25,26,28]
	- poor/unreliable internet	[8,9,14,17]
Organizations, policies, legislation	- lack of integrated care policies	[7,11,13,26]
	- financial barriers: financial concerns, financial constraints	[8,17,20,22]
	- lack of trust between organizations, data sharing	[9,11,15,19,20,22,23,27,28]

**Table 2 jpm-12-00467-t002:** Overview of the results—facilitators.

	Facilitating Factors for eHealth and mHealth Implementation in Chronic Diseases	References
Individual	- motivation to change lifestyle change, learning about health	[7,8,9,12,22,24]
	- improving self-management skills, self-health monitoring	[7,8,14,15,16,17,19,21,23,24,25]
	- reducing the number of hospitalisations	[16,22]
	- eHealth and mHealth saving time	[7,20,22]
	- suport from family and/or caregivers	[17,18,22,23]
	- increase in physical activity and mental stimulation	[16,22]
	- improved connection and communication with physician	[7,8,11,15,16,17,20]
Technological	- globally standardised coding schemes	[28]
	- easy to use eHealth software	[8,12,17,22,24,27]
	- ability to use eHealth applications via mobile devices	[20,22,26]
Organizations, policies, legislation	- implementation of international legislation, e.g., GDRP and Directive 95/46/EC	[28]

**Table 3 jpm-12-00467-t003:** Expected barriers and facilitators for the implementation of the Quadruple Aim [29].

Quadruple Aim	Barriers for the Implementation of Personalised Medicine Interventions	Facilitators of the Implementation of Personalised Medicine Interventions
**improving the individual experience of care**	- lack of awareness of PM services - lack of skills of elderly people	- increased number of training sessions/conferences showing the possibility of PM - communications and informing citizens of the benefits of PM
**improving the health of populations**	- mainly specialised and service-centred, rather than patient-centred - lack of a user-friendly technology - access to individual data at the same time, guaranteeing their security - medical digital solutions are overly fragmented due to national legislations derogating GDPR/national evaluation - conflicts between regional and national competencies	- diffusion of patient-centred approaches - availability of personalised data as the basis for a decision for a personalised diagnosis and treatment
**reducing the per capita cost of healthcare**	- lack of financial incentives provided to HCPs to experiment with such solutions - some managed care executives feel that PM will increase the cost of prescription medicines	- mutual recognition for medical digital solutions published in other EU member states - centralised evaluation system and transparency between reimbursement rates of national healthcare systems
**improving the experience of providing care (the importance of physicians, nurses and all employees finding joy and meaning in their work)**	- lack of training for healthcare staff - lack of investments in healthcare	- patient advocates and cooperation with researchers and open-minded physicians - healthcare providers that can provide direct contact with patients and explain better the benefits of a PM treatment to them

**Table 4 jpm-12-00467-t004:** Existing solutions in the field of personalised medicine—a synthesis based on the Best Practices Booklet within the Regions4PerMed project [2,30,31,32].

Project Initiative Title	Country	Key National Solutions
Return of genomic data to biobank participants, personalised medicine pilot projects in Estonia	Estonia	The aim of the pilot projects supported by the Estonian Research Agency RITA: development and gradual implementation of the rules, procedures, and principles necessary for the introduction of personalised medicine for general practitioners and specialists. During the project, more than 2000 biobank participants received genetic feedback and were further researched and treated by primary care physicians, oncologists, and medical geneticists as needed. Project website: https://genomics.ut.ee/en (accessed on 19 February 2022) [30]
Onkolotse (Cancer guide)	Germany	Improve personal support for cancer patients and their families along the treatment pathway and across all medical settings (idea: one face to the patient). Onkolotse will help patients and their families to find their personal path through cancer treatment, become informed patients, improve treatment adherence and coping, and help them live with the disease and make the most of their lives. Project website: www.nweurope.eu/codex4smes (accessed on 19 February 2022) [31,32]
How the Techforlife cluster can support and improve Lombardy’s healthcare system	Italy	The main challenge is to use technology to provide a high standard of care in the daily lives of people with chronic diseases; in particular, ensuring a personalised motor and cognitive rehabilitation process while improving the quality of life of patients. The goal is to focus on the monitoring and safety of patients in their homes, to create technological innovation through a high-level scientific approach that is fostered by multidisciplinarity and technology transfer, including the implementation of efficient business models. Project website: https://cluster.techforlife.it (accessed on 19 February 2022) [30]
Organisational and digital development in the care of chronically ill patients	Italy	ASST Vimercate has a defined process that enables the proactive “care” of chronic and frail patients, including the definition of different professional roles, the introduction of an outsourced service centre to manage care activities, the introduction of the role of “case manager” to ensure the quality of the process and care. In addition, ASST Vimercate has begun to use “Big Data Analytics” technologies to develop predictive algorithms that help professionals in the early detection of patients with certain chronic diseases and the occurrence of complications. Project website: https://www.asst-brianza.it/web/ (accessed on 19 February 2022) [30]
Virtual coach and chatbot interactions for cognitive enhancement	Italy	The main challenge is to provide outcome-based integrated care to older people to improve their quality of life and that of their families, while making European health and social care systems more sustainable, and to build an integrated care system between health and social services by proposing an app-based platform that connects informal and formal caregivers and supports with health empowerment through a virtual coach. Project website: https://www.israa.it/home-europa (accessed on 19 February 2022) [2]
BIOCAM—AI approach to doctor’s workload reduction	Poland	BIOCAM is a start-up company developing innovative capsule endoscopy that enhances patients’ comfort of life and provides new healthcare solutions. The most common diseases that can be screened with capsule endoscopy are Crohn’s disease, celiac disease, small bowel tumors, and anemia of unexplained cause. The endoscopy capsule is only 11 mm wide and 23 mm long. Project website: https://biocam.pl/ (accessed on 19 February 2022) [2]
Cardiomatis	Poland	The cloud tool speeds diagnosis and increases efficiency for cardiologists, clinicians, and other healthcare professionals in interpreting ECGs, automating the detection and analysis of about 20 cardiac abnormalities. The software integrates with more than 25 ECG monitoring devices and offers an advanced cloud software interface as a differentiator from traditional medical software. Project website: https://cardiomatics.com/ (accessed on 19 February 2022) [2]
Glucoactive—control diabetes, everywhere, always	Poland	Innovative technology enables non-invasive, automatic measurement of blood glucose levels. The proposed telemedicine solutions, including online storage, enable fast and accurate measurement, as well as features known from premium-class smartwatches. The devices have no replaceable elements such as strips or sensors, they are a one-time purchase, which means cost savings compared to invasive devices. Project website: www.gluco-active.com (accessed on 19 February 2022) [2]
Infermedica	Poland	Infermedica is developing its diagnostic engine to collect admissions, verify symptoms, and guide patients to the right treatment. The company uses artificial intelligence and machine learning to evaluate symptoms and find patterns in the data. The medical team reviews every piece of information added to the medical database to ensure patients receive safe and reliable recommendations. Infermedica develops mobile, web, and chatbot apps that are easy to use and integrate. Project website: https://infermedica.com/ (accessed on 19 February 2022) [2]
Patient Rescue Support Project Wrist-Band Device	Poland	The aim is to develop an innovative care concept tailored to solving the problems associated with demographic change. The wrist-band device can work in two ways: 1. It collects basic data (real-time physiological signals) on the wrist to help a person with frailty, without having to call the emergency services. The data from the wristband ID is transmitted to the dispatcher; 2. It can give doctors and medical staff access to clinic data. Project website: http://www.wrp.info.pl (accessed on 19 February 2022) [2]
StethoMe^®^	Poland	The company has developed an intelligent solution to improve diagnosis in primary care. StethoME is an AI-powered healthcare solution that enables automated and remote lung and heart exams. It provides the telemedicine solution with the missing piece of the puzzle of remote interaction between the professional, the physician, and the patients themselves. Project website: https://stethome.com/ (accessed on 19 February 2022) [2]
HAITool—A real-time hospital infection surveillance and hospital-wide intelligent clinical decision support system	Portugal	To solve the problem of multiple sources of hospital/patient data, a web-based information system was developed that supports an SQL server that extracts and summarises patient data, microbiology laboratory results, and pharmacy data. Data are extracted at regular intervals from hospitals’ existing information systems by an ExtracteTransformationeLoad (ETL) module, using intelligent automated routines and then being processed and aggregated in a single data warehouse. Project website: http://www.haitool.ihmt.unl.pt (accessed on 19 February 2022) [30]
NAGEN 1000: An example of a project for regional implementation of personalized genomic medicine in healthcare	Spain	“NAGEN 1000” is a pilot study to integrate recent advances in modern genomic technology into clinical practice. The study mainly targets patients with rare diseases and their families. The whole-genome data provide answers not only for rare diseases, but also for the field of personalised prevention by analysing genetic factors associated with the risk of serious, preventable diseases. In addition, the analysis of pharmacogenomic variants provides initial insights into the type and dosage of certain drugs that are tolerated by individuals. Project website: https://www.nagen1000navarra.es (accessed on 19 February 2022) [30]

## Data Availability

Not applicable.

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
