# Peer review of "eHealth and mHealth in Chronic Diseases—Identification of Barriers, Existing Solutions, and Promoters Based on a Survey of EU Stakeholders Involved in Regions4PerMed (H2020)"

_jpm, 2022, doi:10.3390/jpm12030467_

Round 1

Reviewer 1 Report

Previously I've commented that obtaining useful information regarding the barriers and facilitators for e-health and m-health implementation is relevant. Gathering information from several countries in Europe, and from experts matters for the definition of policies, nevertheless, the document still doesn't show the survey results in the way they can be analyzed by the reader, there's also no information with details about the conference review is inadequate. Regarding the first submission, there are some improvements although the main questions remain. On the other side, particularly the new text has some problems regarding the English language and there are also some typing mistakes. Eventually, the title could express better that the document is based on a conference, and perhaps that change could clarify the intent and explain the author choices

Author Response

We have revised the text to address your concerns and hope it is now clearer. Typographical errors have been revised.
We would like to note that only the group of respondents was based on the speakers at the conference, while the document itself is not based solely on the conference as a source. This was explained in more detail in the methodology

Reviewer 2 Report

The purpose of the manuscript is to provide a review of the barrier and facilities of eHealth and mHealth in Chronic diseases. The authors claimed to have a conclusion that based on data collected from the surveys: barriers to the implementation of eHealth and mHealth are mainly lack of skills of senior citizens as well as lack of user-friendly technologies. Although the scope of the review paper is very much appropriate and time sensitive, but the manuscript is very weak in several significant aspects. Such as: very little literature survey, lacks organization, too many direct copying from other paper, lacks analysis/comparison and lacks any fruitful outcome/future directions.  In my opinion, the manuscript is not suitable for publication in the “Journal of Personalized Medicine”.

Major points:

  1. While discussing section 5.4 the authors only included two reference which is very insufficient and proves serious lack of literature survey. Authors need to do extensive literature survey.
  2. There need to several table which should be populated with existing products as well as current research involving eHealth and mHealth. There are plenty of articles and review papers available right now.
  3. The manuscript seriously lacks current state of art.
  4. Although author mentioned some limitations in conclusion but there is no concrete proposal to overcome those issues. Although several review papers are already published on these topics the Author need to elaborate to make their manuscript better or improve over those papers.
  5. Line 45,46,47 is copied from https://www.i-jmr.org/2016/1/e5/. Even while using citation, you are not allowed to directly copy from anyplace.
  6. Line 85-96 is copied from https://obc.opole.pl/Content/15274/PDF/MSP2020_02.pdf
  7. The number of participants given in section 3.3 is not sufficient to conclude any significant decision. Authors need to explain how they have come to conclusion that the number of participants is sufficient?
  8. No reference given for “data collected”, “existing article”, “online surveys” written in section 3.4
  9. What is meant by “general information” in section 3.6.1?
  10. What were the questions in 3.6.2? any example? How to justify that those questions were standard questions?
  11. There are way more than 15 articles available which should be included in quantitative synthesis. The authors have missed significant number of articles which should have been included. Please use google scholar to find out more relevant articles and increase the number from 15.
  12. Please explain why only PubMed was used.
  13. Line 173,174,175 copied from https://bmcpublichealth.biomedcentral.com/track/pdf/10.1186/s12889-021-11623-w.pdf
  14. Line 194-202 is copied from https://www.i-jmr.org/2016/1/e5/
  15. Line 208,209,210 is copied from https://obc.opole.pl/Content/15274/PDF/MSP2020_02.pdf
  16. Line 226-229 is copied from https://pubmed.ncbi.nlm.nih.gov/32628118/
  17. Line 235-238 is copied
  18. Line 243-265 is copied

Minor points:

  1. In line 58 they mentioned about “many pilot projects” but no reference provided. Again, in line 60 there’s mention of “many initiatives” but no reference.

Need extensive literature survey and better planning. Plenty of review papers are available from good publishers about mHealth and eHealth. Authors need to compare currently available device, research, techniques and point out the limitations and future directions. Also, the authors need to refrain from copying directly from other journal or papers.

Author Response

COMMENTS

AUTHORS’ REPLY

While discussing section 5.4 the authors only included two reference which is very insufficient and proves serious lack of literature survey. Authors need to do extensive literature survey.

We thank you for this comment and agree with the reviewer that further elaboration on this point would be helpful. We have done a more extensive literature search.

There need to several table which should be populated with existing products as well as current research involving eHealth and mHealth. There are plenty of articles and review papers available right now.

We have added a table of existing products and a table of the Quadruple Aim of integrated care.

Please see pages 6-7 of the revised manuscript, line 210 - 211, pages 7-8, line 224 and pages 9-13, line 295.

The manuscript seriously lacks current state of art.

We have revised the text to address your concerns and hope it is now clearer.

Although author mentioned some limitations in conclusion but there is no concrete proposal to overcome those issues. Although several review papers are already published on these topics the Author need to elaborate to make their manuscript better or improve over those papers.

We have revised the text to address your concerns and hope it is now clearer.

The number of participants given in section 3.3 is not sufficient to conclude any significant decision. Authors need to explain how they have come to conclusion that the number of participants is sufficient?

The participants are experts in the field of eHealth, mobile health services and personalized medicine in general. They are associated with research institutes, private funders, policy makers, etc.

As mentioned above, the questionnaires were sent to the stakeholders of the Regions4PerMed (Interregional Coordination for Rapid and Deep Adoption of Personalized Medicine) project – speakers at conferences and workshops, participants of these events.

Explanations were added to the manuscript in the limitations section.
 Please see page 19 of the revised manuscript lines 588-606.

No reference given for “data collected”, “existing article”, “online surveys” written in section 3.4

References were added, please see page 4 lines 149-154 of the revised manuscript.

What is meant by “general information” in section 3.6.1?

Gender and nationality, more clearly explained in the text. Please see page 3 of the revised manuscript, line 94.

What were the questions in 3.6.2? any example? How to justify that those questions were standard questions?

We have added an example of the question. Please see page 4 of the revised manuscriptlines 172-176. The questions are from the author’s questionnaire, which was approved by the supervisor (professor with experience in qualitative and quantitative studies). Explanations were added to the manuscript in the limitations section.

Please see page 19 of the revised manuscript  lines 598-606.

There are way more than 15 articles available which should be included in quantitative synthesis. The authors have missed significant number of articles which should have been included. Please use google scholar to find out more relevant articles and increase the number from 15.

The way articles are selected in Pubmed has been changed.

Medline and Academic Search Ultimate articles have replaced Pubmed articles.

Please see page 3 of the revised manuscript line 117-124 and pages 4-5 of the revised article line 184-205.

Please explain why only PubMed was used.

The way articles are selected in Pubmed has been changed.

Medline and Academic Search Ultimate articles have replaced Pubmed articles.

Please see page 3 of the revised manuscript lines 117-124 and page 4-5 of the revised article lines 184-205.

Authors need to refrain from copying directly from other journal or papers

It has been revised. We apologize for our error.

Please see lines 55-59, 105-116, 298-303, 338-353, 365-368, 386-389, and 410-424 of the revised manuscript.

In line 58 they mentioned about “many pilot projects” but no reference provided. Again, in line 60 there’s mention of “many initiatives” but no reference.

It has been revised. We apologize for our error

Need extensive literature survey and better planning. Plenty of review papers are available from good publishers about mHealth and eHealth. Authors need to compare currently available device, research, techniques and point out the limitations and future directions

We have placed more emphasis on an extensive literature review throughout the text.

Round 2

Reviewer 2 Report

It is evident that authors have worked really hard and made a significant amount of changes to their manuscripts.

The only point I can raise now is they need to reduce the amount of information directly used from this article (in table 4): https://www.regions4permed.eu/wp-content/uploads/2020/07/KA1_Best-Practices.pdf. 
they need to do a better job at rephrasing or summarizing. Also, they have not provided references of that pdf inside the table. 

Author Response

Thank you for this comment and we agree that the amount of information needs to be reduced. We have also provided references in Table 4.

This manuscript is a resubmission of an earlier submission. The following is a list of the peer review reports and author responses from that submission.

Round 1

Reviewer 1 Report

Obtaining useful information regarding the barriers and facilitators for e-health and m-health implementation is relevant. Gathering information from several countries in Europe, and from experts matters for the definition of policies, nevertheless, the document doesn't show the survey results in the way they can be analyzed by the reader, there's also no information with details about the conference and even the systematic review is inadequate. Besides that, very little is told regarding how country and people surveyed were selected and who they represent. In the end it's not understandable the surveys role for the manuscript.

Reviewer 2 Report

Dear Authors,

electronic and mobile medicine in chronic diseases is a very interesting and currently fashionable topic, especially during this pandemic period (where, for example, telemedicine has been used to avoid the spread of COVID). Unfortunately, I think that the paper is not suitable for publication in this Journal.

The type of article is not the most suitable given the poor structure and repetitiveness of the manuscript. The text does not add any novelty to the literature: it could be an Editorial or Letter rather than an Original Article.

The methods of acquiring the survey (including the criteria for selecting the responders and the various questions) are not specifically detailed. The systematic review does not meet the minimum publication criteria (i.e., checklist PRISMA). There are many abbreviations, especially in the results, which are not understandable. The references are also disordered (especially the numbering) and often not authoritative.

Sorry, but given these observations, I think that this paper cannot be published.